# Metabolic model-based ecological modeling for probiotic design

James D Brunner[1,2]*, Nicholas Chia[3]*

[1]Biosciences Division, Los Alamos National Laboratory, Los Alamos, United States; [2]Center for Nonlinear Studies, Los Alamos National Laboratory, Los Alamos, United States; [3]Data Science and Learning, Argonne National Laboratory, Lemont, United States

**Abstract** The microbial community composition in the human gut has a profound effect on human health. This observation has lead to extensive use of microbiome therapies, including over-the-counter 'probiotic' treatments intended to alter the composition of the microbiome. Despite so much promise and commercial interest, the factors that contribute to the success or failure of microbiome-targeted treatments remain unclear. We investigate the biotic interactions that lead to successful engraftment of a novel bacterial strain introduced to the microbiome as in probiotic treatments. We use pairwise genome-scale metabolic modeling with a generalized resource allocation constraint to build a network of interactions between taxa that appear in an experimental engraftment study. We create induced sub-graphs using the taxa present in individual samples and assess the likelihood of invader engraftment based on network structure. To do so, we use a generalized Lotka-Volterra model, which we show has strong ability to predict if a particular invader or probiotic will successfully engraft into an individual's microbiome. Furthermore, we show that the mechanistic nature of the model is useful for revealing which microbe-microbe interactions potentially drive engraftment.

*For correspondence:
jdbrunner@lanl.gov (JDB);
chia@anl.gov (NC)

## Editor's evaluation

This manuscript uses genome-scale metabolic modeling to estimate interspecies interactions and subsequently assess engraftment outcomes. This is an important line of work with potentially broad applications in different fields, including microbiota studies. The authors provide solid evidence to support the usefulness of their proposed approach in engraftment studies.

## Introduction

Microbiome research has come to encompass key areas of disease, ranging from infections (*Antharam et al., 2013*; *Honda and Littman, 2012*; *Battaglioli et al., 2018*) and cancer prevention (*Moss and Blaser, 2005*; *Walther-António et al., 2016*; *Kim et al., 2020*) to systemic immune and neurological responses (*Severance et al., 2016*; *Kang et al., 2014*; *Chen et al., 2016*). The effect of the microbiome on health is now undeniable, and every year in the US over 400,000 people collectively spend $1 billion dollars on over-the-counter probiotics intended to alter their microbiome (*Kristensen et al., 2016*). Many of the purported interactions between microbes and health involve *resident* microbiota and their interactions with the host, i.e., the interface between microbial ecology and human health. The goal of microbiome-targeted interventions is therefore to promote health by 'restoring and maintaining the microbiota and the crucial health-associated ecosystem services that it provides' (*Costello et al., 2012*).

Despite the many links between the microbiome and health, our ability to deploy probiotics to modify the microbiome as intended has been met with relatively little success (*Mullard, 2016*; *Zhu et al., 2019*; *Yuan et al., 2017*; *Zhao et al., 2021*; *Wu et al., 2017*; *Lawson et al., 2019*). Studies looking at the ecological effects of probiotic administration show that administration of a probiotic is not sufficient to alter the community in the desired way. Specifically, engraftment of the adminis- tered microbial species is often limited, with only one-third to one-half of patients showing any signs of medium- or long-term engraftment (*Maldonado-Gómez et al., 2016*; *Pudgar et al., 2021*). We offer the argument that probiotic interventions are primarily ecological in nature; their purpose is to reshape the complex microbial communities in our body in beneficial ways. Therefore, to predict whether a probiotic has the desired effects in the gut microbial community, we need more studies examining the ecology of probiotic interventions (*Walter et al., 2018*; *van den Berg et al., 2022*). A mechanistic, personalized approach to probiotic design—one rooted in empirical metabolic data and ecological principles—has the potential to propel the field forward.

Previous work trying to predict engraftment has been mostly restricted to fecal microbiome trans- plant (FMT) studies using non-mechanistic classifiers (*Smillie et al., 2018*; *Podlesny et al., 2021*). While potentially predictive, such classifier approaches are sensitive to the underlying assumptions or conditions in which the study is carried out. Because such models are built using statistical methods on data that is assumed to be uniformly collected, these models cannot be generalized to new circum- stances. It would be unexpected to see predictions built from patients with diarrhea, undergoing bowel prep, taking antibiotics, and given an FMT accurately predict what happens in patients taking orally administered probiotics. For these reasons, more detailed mechanistic approaches, such as genome-scale metabolic modeling, have been recently explored as more generalizable alternatives (*Heinken et al., 2021a*; *Heinken et al., 2021b*; *Dillard et al., 2021*; *Jenior et al., 2021*).

The goal of this work is to examine the use of metabolic modeling-informed population dynamic approaches. This builds on top of related work in both constraint-based metabolic modeling and population models such as Lotka-Volterra (LV). It is worth highlighting that the use of dynamic flux balance analysis (FBA) for population models has also been a well-published approach. Despite these successes, there are a number of practical drawbacks for communities of high complexity such as labor-intensive interpretation and high computational complexity.

Population models such as the generalized LV (gLV) model are popular tools for understanding micro- bial community dynamics in a mechanistic manner (*Stein et al., 2013*; *Friedman et al., 2017*; *Angulo et al., 2019*; *Kuntal et al., 2019*). However, these models are in general difficult to parameterize, with state-of-the-art gradient-matching procedures requiring somewhat dense time-longitudinal data with many replicates (*Bucci et al., 2016*). Furthermore, it has previously been shown that parameters fit from data to these models do not extend to novel environmental situations, and may even change with the addition of a new taxa to the community (*Brunner and Chia, 2019*; *Momeni et al., 2017*). These drawbacks make such mechanistic population models impractical for predicting engraftment. However, by leveraging metabolic modeling we are able to parameterize population models in a way that can be easily adapted to novel environments and does not require dense time-longitudinal data. This allows us to use these models to predict microbial engraftment into a community. See *Figure 1* for a comparison of our method with standard parameter fitting.

Genome-scale metabolic models (GSMs) encode the metabolic pathways available in a cell, allowing simulation of cellular metabolism and growth (*Lewis et al., 2012*). These tools have been used extensively to understand and engineer microbial mono-cultures, and have recently begun to be used to understand and predict the composition and metabolic function of microbial communities (*Chan et al., 2017*; *Diener et al., 2020*; *Kim et al., 2022*). Community methods using GSMs often suffer from a number of drawbacks. These can include a focus on equilibrium states, high complexity and computational cost, and the difficulty of providing accurate GSMs for each community member. By layering simple, pairwise GSM community modeling with population models, we allow for predic- tion that includes transient behavior (e.g. how quickly an extinction happens within a community) and reduces computational complexity. The availability of accurate GSMs remains a major difficulty, but we demonstrate that progress can be made using algorithmically generated models from whole genomes.

In this paper, we present a method to predict engraftment of an invader into a microbial commu- nity in the following manner. First, we construct an interaction network of the microbial taxa found

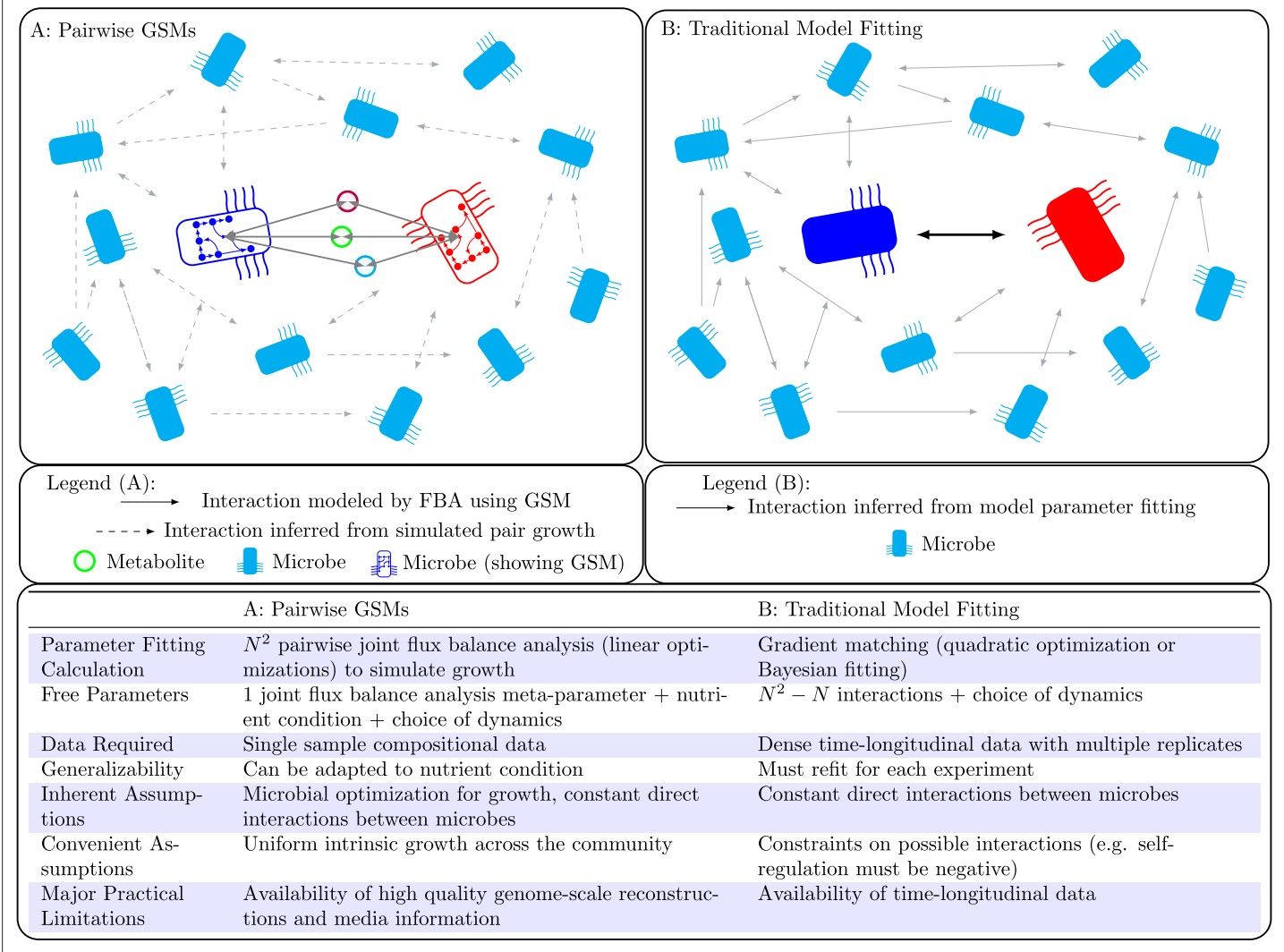

**Figure 1.** Population models such as Lotka-Volterra generally require dense time-longitudinal data to accurately parameterize, and directly fit interactions between species (solid arrow in right panel). In this work, we leverage genome-scale metabolic modeling to parameterize population models with only genomic data from a single time-point. This is accomplished by modeling microbial interactions with their shared environment (colored circles in the left panel).

in a sample from the community using pairwise FBA with resource allocation constraints (**Kim et al., 2022**), which requires a GSM for each taxa. To construct the necessary GSMs, we matched taxa found in the sample to the genomes found in the RefSeq database (**O'Leary et al., 2016**), and used these genomes to build models using ModelSEED (**Seaver et al., 2021**). A complete list of the genomes used can be found in **Supplementary file 1**. We then simulated pairwise growth using community FBA with growth media adapted from the AGORA genome-scale modeling project (**Heinken et al., 2023**); precise details are available in *S1 Data*.

After network construction, we make a prediction based off of simulations with the gLV model (**Edelstein-Keshet, 2005**; **Stein et al., 2013**; **Friedman et al., 2017**), and test the model's predictive potential by predicting the outcomes of microbiome invasion experiments (**Maldonado-Gómez et al., 2016**) from the initial presence/absence of species in each sample. The LV model is widely used, but we found that this model could lead to uncontrolled simulated growth. We therefore also investigated alterations to the LV model that dampen the numerical instability.

Despite the drawbacks to both population models and GSMs, limitations in our ability to construct high-quality GSMs, and the limited data requirement of our method, our method produces positive predictive value in the scenario in which it was tested. Our method therefore represents a 'low-budget'

**Table 1.** Area under the receiver operating characteristic curves for our method's predictions of 22 samples from each of two time-points (TP) using six sets of parameters inferred from joint flux balance analysis (FBA) with six different sets of hyperparameters.

The first three sets of inferred parameters differ in the 'resource allocation constraint (RAC)' in joint FBA. We used values of 35 and 70 for this parameter, as well as using joint FBA without RAC. The next three sets of parameters were inferred using an RAC value of 35 but changes to the model environments. EU average diet (C halved/doubled) had the major carbon sources of the 'EU average diet' (D-maltose, sucrose, D-fructose, and D-glucose) halved or doubled in availability, and 'complete medium' simulated the availability of any exchangeable metabolite at uniform simulated inflow.

| | Baseline TP (p-value) | Treatment TP (p-value) |
|---|---|---|
| EU average diet (RAC 35) | 0.6161 (0.1020) | 0.8482 (<0.001) |
| EU average diet (No RAC) | 0.6161 (0.1020) | 0.8571 (<0.001) |
| EU average diet (RAC 70) | 0.6429 (0.0741) | 0.8482 (<0.001) |
| EU average diet (C halved) | 0.6071 (0.1107) | 0.8393 (<0.001) |
| EU average diet (C doubled) | 0.6339 (0.0808) | 0.8304 (0.0010) |
| Complete medium | 0.6071 (0.1155) | 0.7143 (0.0221) |

tool that nevertheless can be useful. This success is also evidence that more complex methods that expand on our simple approach could provide great predictive value with a still limited data requirement. Furthermore, we perform two types of sensitivity analysis to demonstrate that the mechanistic nature of the model provides additional insight into the impact of the various components of the network. We perform simulated knock-out experiments to test how sensitive engraftment is to each community member, and we use parameter sensitivity analysis to test how sensitive engraftment is to each network connection.

## Results

### Predictive value of the method

We first examine the ability of our metabolic modeling-based approach to successfully predict engraftment versus non-engraftment for different microbial species introduced orally across different experimental or clinical trial settings. The study that we use to test predictive power, authored by *Maldonado-Gómez et al., 2016*, involved the introduction of candidate probiotic into an established microbial community. In the study, *Bifidobacterium longum* AH1206 was administered as an oral probiotic to 23 subjects, with data available for 22 of these subjects. We predicted engraftment of the candidate probiotic using the 'baseline' samples, which were taken before introduction of the probiotic, and the 'treatment' samples, which were taken during probiotic administration. We compared our predictions to a binary 'engrafter' or 'non-engrafter' classification based on cell culture at later time-points.

As a metric of classification success for the data set, we use the area under the curve of the receiver operator characteristic (AUC-ROC). This metric provides a measure of performance based on the model's ability to identify true positives while avoiding false positives, so that 1 is perfect classifier performance and o.5 is equivalent to random classification (i.e. flipping a coin for each sample). We used six sets of parameters inferred from joint FBA with different hyperparameters to parameterize the gLV model, and report the resulting predictive value in *Table 1*.

Our method showed moderate positive predictive value on predictions from the baseline samples, and good predictive value on predictions made from samples taken during treatment. Improved prediction between baseline and treatment samples suggests that a change in microbiome as a response to the introduction of the probiotic impacts our method. This in turn suggests that our method captures at least part of the underlying biological processes that determine engraftment. Furthermore, these

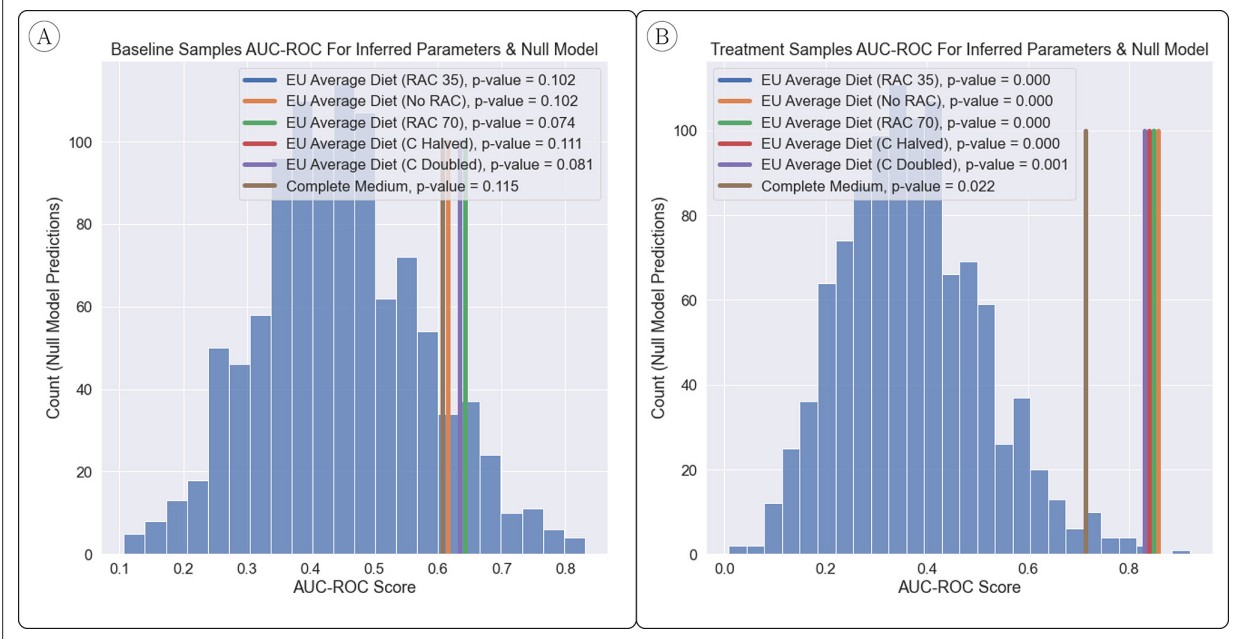

**Figure 2.** Comparison of AUC-ROC Between Inferred Parameters & Null Model. (**A, B**) The power to predict engraftment versus non-engraftment of the *B. longum* probiotic was relatively robust to the joint flux balance analysis (FBA) hyperparameter setup, as shown and measured by the area under the receiver operator characteristic curve (AUC-ROC) (with 1 being perfect classification of engraftment and 0.5 being random). The null model had generally less predictive power than inferred parameters, in particular when predictions were made with samples from the 'treatment' time-point. The overall improvement in predictions for the 'treatment' time-point vs. the 'baseline' time-point suggests that early changes to the microbiome after introduction of a probiotic play a significant role in determining eventual engraftment. p-Values were estimated based on over 1000 samples from a null model.

results were mostly robust to choice of hyperparameters, with the exception being that the complete medium showed significantly worse predictive value on the samples taken during treatment.

In *Figure 2*, we show the estimated significance of our results when compared to the null model (720 samples of the null model). We use an approximate 'edge swapping' procedure as a null model, which draws interaction parameters randomly from the set of inferred interaction parameters, with replacement. In other words, our null model shuffles the original set of interaction parameters so that any $\beta_{ji}^N$ in the null model is some $\beta_{ji}$ in the inferred parameter set. This ensures that the network of interactions in our null model has the same edge weight distribution as that of our inferred model so that we may test the significance of the interactions inferred, rather than summary statistics such as diversity.

We also compared our method with two standard machine learning techniques: a support vector machine (SVM) and a random forest (RF) classifier. Unlike our method, which does not use known classifications to 'learn' a model, both of these techniques require a 'training' set of data in which samples are labeled according to their known classification. We trained these models on 16 of the 22 samples (~70% of the data), and tested their ability to predict the remaining 6 samples. We repeated this procedure 1000 times with randomly chosen testing/training sets. For both classifiers, we used the relative abundance data of the set of taxa that our model considered, as well as binarized presence/absence version of the data. We did this because our method only considers presence or absence of taxa, rather than relative abundance. The SVM classifier did not outperform random assignment, while the RF classifier performed, on average across train/test splits, about as well as our method. Our method, therefore, provides the same ability to classify unknown samples as an RF classifier without the need for any a priori known sample classifications (*Figure 3*).

## Uniform shifts in interaction parameters

One consequence of using the gLV model in our method is the possibility of finite-time blow-up in simulation. This happens when one or more taxa approach infinite biomass in finite simulation time. In practice,

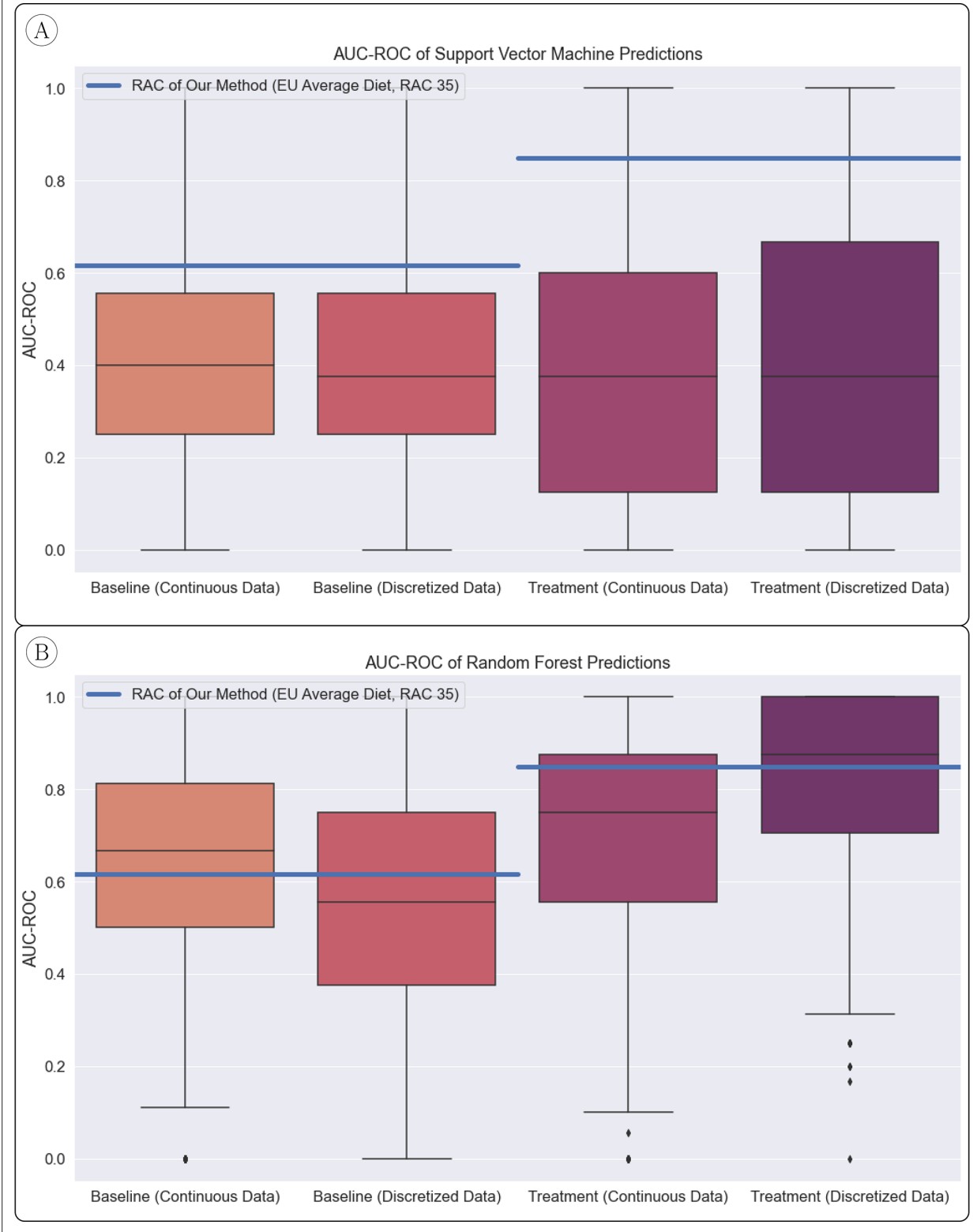

**Figure 3.** AUC-ROC of Standard Classifier Predictions. (**A**) Our method (horizontal lines) significantly outperformed the support vector machine classifier, which was assessed with 1000 random train/test splits. (**B**) The random forest classifier, also assessed with 1000 train/test splits, performed similarly on average to our method.

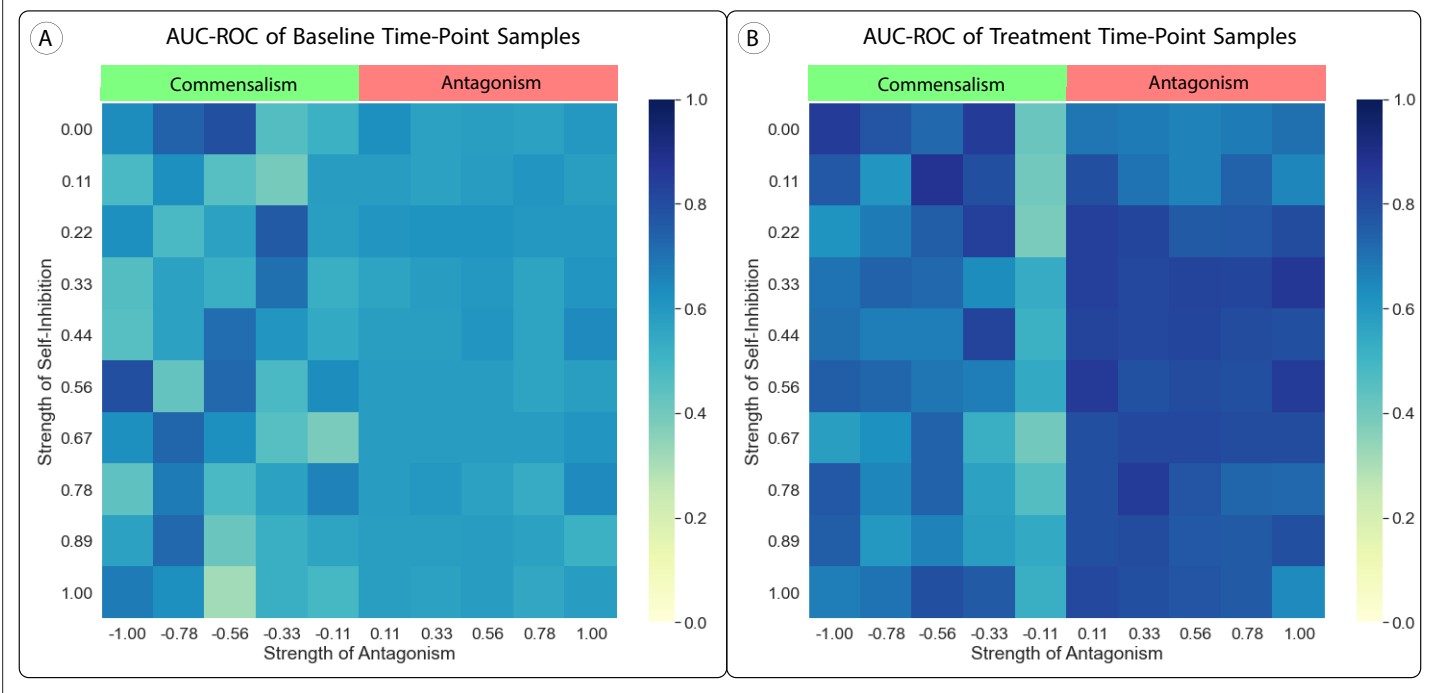

**Figure 4.** Effect of Uniform Parameter Shifts in the Lotka-Volterra Model. (**A, B**) We altered the generalized Lotka-Volterra model with uniform shifts in parameters which added either antagonism or self-inhibition to the model. We tested self-inhibition with values from 0 to 1 (no self-promotion) and antagonism with values from 1 (complete antagonism) to –1 (complete commensalism), with all $\beta_{ij} \in [0, 1]$. Neither change had a significant impact on the area under the receiver operator characteristic curve (AUC-ROC) of our method's predictions, although adding commensalism to the model made the model and resulting predictions much less stable due to increased finite-time blow-up.

whenever one or more taxa increase to a level much higher than the initial total biomass of the simulated system, simulation is slowed down and results may be less reliable. There are two simple methods to prevent this from happening. The first is to alter every interaction parameter to be more negative (see *Equation 3*), making the system uniformly more antagonistic and less likely to show finite-time blow-up. The second is to include negative self-inhibition (e.g. add $\beta_{ii} < 0$ parameters [see *Equation 4*]).

We tested both of these alterations by implementing them and computing predictions and the AUC-ROC of those predictions. *Figure 4* shows that these changes did not have a significant impact on the performance of our predictions, as long as only antagonism was considered. When we shifted parameters in the positive direction (simulating commensalism or mutualism), finite-time blow-up was more likely and our results were much less stable.

## Interpreting mechanism—sensitivity to inferred parameters

One major advantage to any mechanistic model is that we may measure the sensitivity of our results to perturbations in the model. Here, we investigate the results of perturbing the model of the *B. longum* experiments in two ways: simulating 'knock-out' experiments and computing sensitivity to changes in interaction strength.

First, we simulate knock-out experiments by removing a taxa from the full network of interactions. As a result, this organism will be removed from any sample it was previously present in. We then measure the effect this has on our prediction of *B. longum*'s likelihood of engrafting based on the LV dynamics.

Next, we measure the sensitivity of *B. longum* growth to each interaction parameter in the LV model. That is, we measure the effect of perturbing each parameter individually on the simulated abundance of *B. longum* at equilibrium according to antagonistic LV dynamics.

### Sensitivity to community members

In order to investigate how sensitive *B. longum* engraftment is to each of the other species present in any sample of the *B. longum* experimental data set, we simulate 'knock-out' experiments and observe

**Table 2.** We experimented with simulated knock-outs for the top 5 taxa in average abundance in the data.

The 'sample proportion' column gives the proportion of samples in the data set that contain the organism that was knocked out. The 'average score difference' is the average effect of the knock-out on our computed engraftment score, with a positive number indicating an average increase in engraftment after knock-out of the organism (implying a negative interaction between the organism and *B. longum)*. The final column shows the impact on our predictions of removing the organism from the analysis.

| | | Sample proportion | Average score difference | AUC-ROC difference |
|---|---|---|---|---|
| Baseline TP | *Bifidobacterium adolescentis* | 0.954545 | 0.010114 | 0.017857 |
| | Uncultured *Ruminococcus* sp. | 1.000000 | 0.012803 | 0.026786 |
| | Uncultured *Clostridium* sp. | 1.000000 | 0.006259 | –0.008929 |
| | *Eubacterium rectale* | 1.000000 | 0.006183 | 0.017857 |
| | *Faecalibacterium prausnitzii* | 1.000000 | –0.002250 | 0.000000 |
| Treatment TP | *B. adolescentis* | 0.954545 | 0.015415 | 0.000000 |
| | Uncultured *Ruminococcus* sp. | 1.000000 | 0.016707 | 0.008929 |
| | Uncultured *Clostridium* sp. | 1.000000 | 0.014424 | 0.017857 |
| | *E. rectale* | 1.000000 | 0.011133 | 0.026786 |
| | *F. prausnitzii* | 0.954545 | 0.013323 | 0.035714 |

the effect this has on our prediction of engraftment. For each simulated knock-out, we removed a taxa from every sample it was present in and repeated our predictive procedure. We recorded the difference in sample score for engraftment of *B. longum* (*Equation 2*) as well as the change in AUC-ROC of our set of predictions. We simulated knock-outs for the five most abundant taxa in the data (averaging across samples).

*Table 2* shows the summary statistics of the simulated knock-out experiments. Of the five knock-outs and two sample time-points that we tested, all but one increased predicted engraftment by a slight amount. Interestingly, most of these also increased prediction accuracy slightly, suggesting that the increase in engraftment was more pronounced among the true engrafter samples.

This experiment demonstrates that incorporating mechanism into prediction can provide useful insight beyond prediction. Because our model considers the network of interactions between microbial taxa, we are able to experiment with the effects of each individual taxa by using simulated experiments like knock-outs.

## Sensitivity to interactions

Our method is based on the network of interactions between microbial species, which ties together the individual interactions between each pair of species into one complex picture. This means that the interaction between two species unrelated to *B. longum* may have an effect on *B. longum*'s growth. We can compute this effect by computing how *B. longum*'s simulated abundance changes as we vary each parameter. Precisely, we can compute the derivative of *B. longum* with respect to each parameter $\beta_{lk}$ in the model.

We computed the sensitivity of our engraftment score to 8 of the interaction parameters in the model, using an RAC value of 35 and the 'EU average diet'. For each of the baseline and treatment time-point sample sets, we chose the 2 strongest negative and 2 strongest positive interactions that appeared in at least half of all samples, as well as the 2 strongest positive and 2 strongest negative

**Table 3.** The average sensitivity of engraftment score across 8 parameters and the 22 baseline and 22 treatment samples, as well as the average (across samples) variance across the 8 edges.
The 8 edges were chosen because they were the 2 strongest positive edges, 2 strongest negative edges, the 2 strongest positive direct edges (i.e. with *B. longum* as a target) and the 2 strongest negative direct edges. Detailed sensitivity results for these 8 edges can be found in *Supplementary file 1*.

|  | Baseline time-point | Treatment time-point |
|---|---|---|
| Variance across setups | 4.270608e-06 | 2.621430e-05 |
| Average sensitivity (8 tested edges) | 3.435107e+33 | 7.504735e+10 |
| Variance of sensitivity (8 tested edges) | 6.250203e+68 | 1.626682e+23 |

interactions that have *B. longum* as a target (and so act directly on the probiotic) that appeared in at least half of all samples. We observed that engraftment score was very sensitive to all the parameters chosen. In *Table 3*, we show the average sensitivity across the 22 baseline and 22 treatment samples and 8 edges, as well as the average (across edges) variance across the samples. We also include the average (across samples) of the variance in engraftment score across the hyperparameter choices that we tested (with different environmental conditions and RAC values) for comparison. We notice that, in contrast to the model interaction parameters $\beta_{lk}$, engraftment score was not particularly sensitive to hyperparameter choices.

## Discussion

Our work examines the scenario where the goal of probiotic intervention is a long-lasting alteration of the resident host microbiota. Given that the links between microbes and health have a basis in observations of the resident human microbiota (*Gupta et al., 2020*), one common assumption is that alterations to resident microbiota would impact health outcomes. Along those lines, we demonstrate that genome-scale metabolic modeling can be used in a simple way to predict the outcome of species invasion experiments with minimal study data required. This suggests that GSMs can provide value in understanding microbial community dynamics from cross-sectional microbiome relative abundance profiling.

Genome-scale metabolic modeling uses genomic data to predict the growth and resource use of a microbial population by considering the entire internal metabolism of the species. This technique can be extended to community modeling in a number of ways (*Zomorrodi and Maranas, 2012*; *Chan et al., 2017*; *Diener et al., 2020*; *Kim et al., 2022*; *Frioux et al., 2020*), all with relatively high levels of complexity. Community dynamics can be inferred from GSMs using a model known as dynamic FBA, which uses a GSM for each taxa in the community to infer growth rates and dynamic resource usage (*Brunner and Chia, 2020*). While dynamic FBA provides a complete picture of community dynamics according to GSMs, it represents a very complex model that is difficult to analyze and simulate, and still contains a set of unknown parameters.

As a limitation, it is worth noting that all GSM-based methods, including our own, are limited to a large extent by the accuracy of the metabolic model reconstructions. Low-quality reconstructions often omit key reactions, or worse, may include reactions that should not have been included. In all cases, this creates the potential for reaction fluxes or interactions that do not reflect reality. Improving the quality of metabolic model reconstruction is the subject of challenging and nuanced research that includes automated pipelines (*Benedict et al., 2014*; *Seaver et al., 2021*; *Faria et al., 2023*) and hand-curation efforts (*Heinken et al., 2023*). These efforts are essential to improve the accuracy of community metabolic modeling, but are beyond the scope of this particular study.

Here, we present a simpler model—a network of emergent interactions between microbes, whose interaction parameters we can approximate from pairwise genome-scale modeling. The gLV model and other similar microbial network models are popular tools for understanding community dynamics (*Friedman et al., 2017*; *Fisher and Mehta, 2014*; *Angulo et al., 2019*; *Bucci and Xavier, 2014*). Such models are difficult to parameterize for a myriad of reasons. Good parameterization requires relatively dense time-longitudinal data of absolute, rather than relative, community abundance (*Bucci*

*et al., 2016*; *Kuntal et al., 2019*). By taking advantage of GSMs, we provide a parameterized network that does not require any time-longitudinal data. In fact, our method as presented here makes use of previously published GSMs or genomes, meaning that only binary presence/absence information is necessary for a prediction. Additionally, GSMs can be built directly from the genomes found in a sample using automated tools such as *CarveMe* (*Machado et al., 2018*) or ModelSEED (*Seaver et al., 2021*), meaning that our method can be extended to include taxa not found in any database.

Another strength of our approach is its interpretability. The *B. longum* engraftment analysis provides us with an example motivation, which would be to identify other partner microbes that could improve the responsiveness to the engraftment of a target probiotic. In this instance, we demonstrate how this might be done by quantifying the impact of five high-abundance microbes on *B. longum* engraftment. We see, for example, that *B. adolescentis* has a negative impact on *B. longum* engraftment (simulated knock-outs increased engraftment score), while *F. prausnitzii* has a positive impact on *B. longum* engraftment. Although time consuming, a complete set of simulated knock-out experiments is is readily possible using our method, which could identify the largest positive and negative relationships between specific taxa and *B. longum* engraftment.

Inspecting the sensitivity of our method's engraftment score revealed that it is very sensitive to the individual interaction parameters $\beta_{lk}$ but quite robust to changes in hyperparameters. This suggests that the method is sensitive to the interactions between microbes, while robust to changes in computational and environmental parameters. It also suggests that individual interactions between microorganisms can have an out-sized impact on the downstream composition of a microbial community. This is consistent with the idea that probiotics may have a role in treating human disease by reorganizing the microbiome through the addition of one or a few species.

Lastly, we note that we have previously shown that species-species interaction modeling, i.e., models built from interactions between microbes, do not capture the complexities of microbial community dynamics that emerge as communities change in composition (*Brunner and Chia, 2019*). Here, we mitigate this shortcoming by determining interaction parameters from pairwise models under specific metabolic conditions, providing limited environmental context to our method. However, it is unlikely that these interactions remain constant as the microbial community manipulates its environment. We conjecture that prediction can be improved by accounting for changes in microbial interaction as the environment changes. In upcoming work (*Brunner et al., 2023*), we demonstrate that dynamic FBA

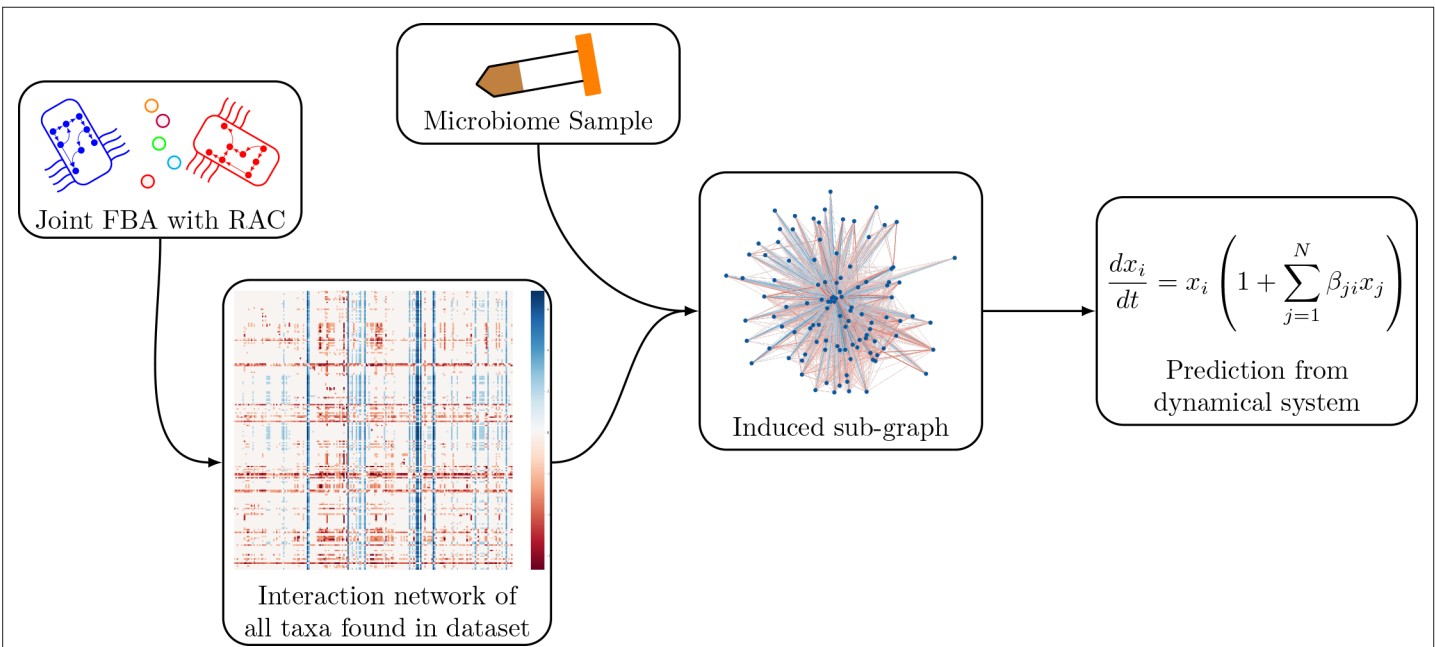

**Figure 5.** Schematic of the modeling process. In brief, we generate an interaction network of genome-scale models using pairwise joint flux balance analysis. To produce a prediction of engraftment for a given sample, we use the taxa present in the sample to generate an induced sub-graph of the full network. This is then used to define the parameters in the generalized Lotka-Volterra dynamical system to generate a prediction of engraftment.

implies that a microbial community behaves according to a discrete sequence of interaction networks over time. Incorporating this dynamic behavior may improve prediction with only a modest increase in model complexity, as long as this sequence of networks can be efficiently determined.

## Methods

Our method, as shown in *Figure 5*, is based on the generalized Lotka-Volterra model. Using genome-scale metabolic models, we infer interaction parameters from pairwise joint flux balance analysis for all pairs of taxa in a dataset. We then predict engraftment of an invader for each sample by simulating the invasion using the generalized Lotka-Volterra dynamics with the taxa present in the sample.

### Assessing receptivity with the gLV model

In order to assess how receptive a sub-graph is to an invading taxa, we used the gLV model, which can be associated with any pairwise graph. We simulated the community represented by each sample to equilibrium and scored the performance of the invading taxa by simulated final abundance or time to extinction.

The gLV model (*Edelstein-Keshet, 2005*) of a community of $N$ species is written as follows:

$$\frac{dx_i}{dt} = x_i \left( 1 + \sum_{j \neq i} \beta_{ji} x_j \right).$$

(1)

Notice that we use a version of this model that assumes an intrinsic growth rate of 1 for each taxa because fitting accurate intrinsic growth rates would require additional data, while we wish to assess the performance of our methods without the need for parameter fitting. Our predictions were based on the equilibrium relative abundance of the target variable (e.g. the variable representing a candidate probiotic) in the dynamical system.

The gLV model can display an array of behaviors, including multi-stability and chaos. For this reason, we use a Monte-Carlo sampling approach to generating predictions, taking repeated random draws of initial conditions and simulating forward. In our experiments, we used 1000 draws (i.e. we compute 1000 separate ODE so) for each sample. In each trial, we scored the network based on the simulated relative abundance of the invader at some large time $T$. It is possible that the invading taxa either dominates the community (i.e. relative abundance approaches 1) or becomes extinct in our simulation. We take this into account by setting a score as follows:

$$s_i = \frac{1}{3} \left( \frac{\text{time to extinction}}{T} + \text{relative abundance at time } T + \frac{\text{time to community dominance}}{T} \right)$$

(2)

for each trial $i$ and averaging over all trials.

To test changes in the structure of the gLV model and control for possible finite-time blow-up of model solutions, we tested two general alterations to the model. The first is a constant shift of the interaction parameters:

$$\frac{dx_i}{dt} = x_i \left( 1 + \sum_{j=1}^{N} (\beta_{ji} - c) x_j \right)$$

(3)

and the second is a shift only in the self-interaction terms:

$$\frac{dx_i}{dt} = x_i \left( 1 - c x_i + \sum_{j \neq i} \beta_{ji} x_j \right).$$

(4)

For both alterations, we used a single constant $c$ across the entire model, and experimented with changes in the value of $c$.

### Inferring interaction parameters

To infer the interaction parameters $\beta_{ji}$ of the dynamical system, we use a network of interactions implied by a technique known as FBA. FBA is a technique used to predict growth rates of microbes

using genome-scale information about their internal metabolisms (*Lewis et al., 2012*). This technique requires a GSM which represents the set of metabolic pathways of a microbe, as well as information about the environmental metabolites available. An optimization problem is then solved to predict a growth rate of the microbe. Similarly, a technique known as 'joint FBA' (*Zomorrodi and Maranas, 2012*; *Chan et al., 2017*; *Kim et al., 2022*) can be used to estimate growth rates of pairs of microbes using a GSM from each microbe.

FBA and joint FBA allow us to simulate singleton and pair growth experiments in silico and use the results of these simulated experiments to parameterize our dynamical model. Simulated biomass when grown in a pair was compared to simulated biomass when grown alone in order to determine an implied interaction between microbes. This interaction is then used as a parameter in the dynamical model, which can be conceptualized as an edge in a network of interactions.

To compute growth simulations, we use joint FBA with a set of resource allocation constraints, which have been shown to improve predictions in pairwise experiments (*Kim et al., 2022*). Briefly, joint FBA with resource allocation constraints solves a linear program defined by the GSMs of the community members. The linear program maximizes total community biomass with the standard FBA constraints as well as the added constraints that each microbe grows a fixed known rate (simulating chemostatic equilibrium), that mass is balanced community wide (with inflow and outflow of nutrients), that there is forced leak of nutrients out of the community (introduced by *Kim et al., 2022* to prevent unrealistically efficient cross-feeding), and that each community member's total internal reaction flux is bounded (the resource allocation constraint). Our problem setup is identical to that described in *Kim et al., 2022*, and we provide Python code to carry out the computation in our supplemental repository.

To perform joint FBA, simulated inflow of nutrients was determined by medium defined as 'EU average diet' by the Virtual Metabolic Human project (*Noronha et al., 2019*; *Ernährungsbericht and Elmadfa, 2012*), with inflow of 10 mmol/day for nutrients not included in the medium but essential for simulated growth. Growth rates of microorganisms were set to be 0.04 g/day, and forced leak of nutrients was set to be 0.1 mmol/day. For our main experiment, we set the RAC constraint to 35 mmol/day as was shown to lead to accurate prediction in Kim et al. We also adjusted this value, as well as the defined medium, in order to assess our method's robustness to meta-parameters.

In order to test the impact of joint FBA hyperparameters including medium and resource allocation on our predictions, we also created networks of interaction parameters by performing joint FBA with five alternative sets of hyperparameters. Two of these networks were made with alterations to the RAC constraint we used in computing joint FBA. We computed interactions using an RAC value of 70, which relaxes the constraint on total flux, and without a resource allocation constraint. The final three networks were constructed with different media, simulating differences in the external environment. We constructed a network with the major carbon sources of the 'EU average diet' halved in availability, and a second with these carbons doubled. The major carbon sources were taken to be D-maltose, sucrose, D-fructose, and D-glucose. Finally, we computed a network with a 'complete medium' in which any exchangeable metabolite for each model was made available with uniform simulated inflow.

We use the log-ratio of simulated growth in pairs and alone as the implied effect of one microbe on another. That is, if simulated growth of species $i$ alone is $x_i$, and growth of species $i$ when coupled in a pair with species $j$ is $x_{ij}$, we weight the edge from species $j$ to species $i$ as

$$w_{ji}^* = \log\left(\frac{x_{ij}}{x_i}\right) \tag{5}$$

We use the log-ratio because it is the simplest choice of translating simulated data to interaction that accounts for fold-changes. Of course, the simple difference in simulated biomass is another possible choice, but this will be much more sensitive to absolute scaling in simulation, so, for example, slow growing taxa will be biased toward smaller interactions in the network. Furthermore, our in silico growth experiments only provide simulated biomass values at chemostatic equilibrium, meaning that fitting LV parameters directly to these data is an underdetermined problem, particularly when pair growth leads to a simulated extinction.

We also re-scale our network so that all edge weights are in the interval $[-1, 1]$, meaning we take as our edge weights

$$w_{ji} = \frac{w_{ji}^*}{\max_{l,k}(|w_{lk}^*|)}. \tag{6}$$

This re-scaling is done for computational convenience, and can be viewed as a time re-scaling of the network dynamics which does not effect our predictions. Additionally, we tested the approach after adjusting either all parameters by addition of a single constant value to every $\beta_{ji}$, as well as by adjusting the parameters by addition of a single constant value to every self-regulatory parameter $\beta_{jj}$.

## Creation of GSMs

In order to carry out our analysis, we needed to construct GSMs for the taxa found in the data. To do this, we matched taxa in the data by NCBI taxa ID to genomes in the NCBI RefSeq database (*O'Leary et al., 2016*). We matched taxa to the nearest (by tree distance according to NCBI's taxonomic database) genome that was listed as 'Chromosome' or 'Complete Genome', or contained at least 1000 genes. On average, this approach allowed us to perform our experiments with ~87–88% coverage of each sample. Taxa for which we failed to match a genome were removed from the data, and all analysis was performed on the remaining. For full details on the coverage of each sample, see *Supplementary file 1*.

Next, we used the ModelSEED database through ModelSEEDPy (*Seaver et al., 2021*) to reconstruct 390 GSMs for each genome that we matched to the data. All models constructed showed positive 391 simulated growth rates using FBA on the 'EU average diet' medium.

## Prediction and evaluation

We used data from *Maldonado-Gómez et al., 2016*, in order to test the predictive power of the method. This data consisted of bacterial community composition of fecal samples taken over the course of experiments in which *B. longum* AH1206 was administered as an oral probiotic to 23 subjects, with data available for 22 of these. Subjects were then differentiated into 'engrafters' and 'non-engrafters' based on the survival of the probiotic strain, as determined by cell culture at later time-points, allowing us to use our method as a classifier. We note that our method 'learns' using joint FBA, and so there is no need to split the data into training and testing sets. In fact, the 'learning' procedure does not make use of any known sample classifications, as a traditional machine learning algorithm would. This allows us to test our predictions on the entire data set, which provides greater confidence in our results. We made and evaluated predictions using two sets of samples, one taken at a 'baseline' time-point before administration of the the probiotic, and one taken at a 'treatment' time-point during administration of the probiotic.

We computed scores for each sample using simulation to time $T = 100$, and varied the discrimination threshold for the binary prediction across the observed values. From this, we computed an ROC curve and its integral (commonly referred to as finding the 'AUC-ROC'). AUC-ROCs take values in the interval [0, 1] and, in general, an AUC-ROC greater than 0.5 indicates positive predictive value of the model. We compared the AUC-ROC to classification using SVM and RF classification (*Pedregosa et al., 2011*). The SVM and RF classifications were performed using both the relative abundances from the data and binary (presence/absence) forms of the data, which matches our method.

We estimated the significance of our predictions against predictions from a null model created by parameterizing an LV system from an 'edge-swapped' full network. The 'edge-swapped' network was constructed by drawing edge weights from the original full network, with replacement, as is commonly done to create a null model in network analysis (*Röttjers et al., 2021*). This approach allows us to create random models that preserve the statistical characteristics of the models that we test on, while changing the specific interactions between taxa. That is, the distribution of the entire set of parameters in the null models $\beta_{ij}^N$ will be approximately the same as the distribution of parameters $\beta_{ij}$ determined by joint FBA. Using this null model ensures that our conclusions are drawn from the joint FBA parameter learning method, and not simply a result of simpler characteristics, e.g., total connectedness of the community. We used 1039 samples from the null model to estimate the significance of our results.

## SVM and RF classification

We compared the performance of our method to the performance of two traditional machine learning approaches: SVM and RF classifiers. We used 1000-fold cross-validation of the predictions made by these tools (i.e. 1000 independent train/test splits). Each trial consisted of splitting the data randomly into 16 training samples and 6 testing samples, fitting both an SVM and an RF classifier to the training samples using standard Python libraries (from scikit-learn v. 0.23.2; *Pedregosa et al., 2011*), and evaluating the 432 model's predictions of the testing samples. This was repeated for both the 'baseline' and 'treatment' sets of samples.

## Sensitivity to community members and interaction parameters

We compute *B. longum* AH1206 engraftment predictions using simulated knock-out experiments simply by removing select taxa from the model. The result is a prediction score computed according to *Equation 2*. This score can be compared to the non-knock-out experiment score to determine if the knock-out increased or decreased the predicted probability of *B. longum* engraftment. We compute and report the relative change in predicted engraftment score as

$$\Delta = \frac{\text{(score with knock-out taxa removed)} - \text{(score with knock-out taxa included)}}{\text{(score with knock-out taxa included)}}. \tag{7}$$

We compute the sensitivity of *B. longum*'s growth to an interaction parameter $\beta_{kl}$ directly by using the chain rule (see, for example, *Zi, 2011*). This has the following form:

$$\frac{\partial}{\partial t}\left(\frac{\partial x_i}{\partial \beta_{kl}}\right) = x_i x_k \delta_{i=l} + x_i \sum_{j \neq i} \beta_{ji} \frac{\partial x_j}{\partial \beta_{kl}} + \left(1 + 2\beta_{ii} x_i + \sum_{j \neq i} \beta_{ji} x_j\right) \frac{\partial x_i}{\partial \beta_{kl}}. \tag{8}$$

*Equation 8* allows us to solve a system of differential equations to determine the sensitivity of invader growth to that parameter. We solve this system of equations to a large time $T = 100$ and report the weighted average of some late-time interval $[t, T]$, with later time-points weighted more heavily, using the final 40 simulation time-points.

## Acknowledgements

The authors would like to thank Dr. Marie Kroeger for her thoughtful comments in helping to prepare this article. JB was supported in this work by the US Department of Energy, Office of Science, Biological and Environmental Research Division using Award number F255LANL2018 and the Los Alamos National Laboratory Center for Nonlinear Studies. This study was partially supported by funds from Mayo Clinic's Center for Individualized Medicine.

## Additional information

### Competing interests

James D Brunner: is an employee of Triad National Security, LLC. Nicholas Chia: is an employee of UChicago Argonne, LLC.

### Funding

| Funder | Grant reference number | Author |
| --- | --- | --- |
| U.S. Department of Energy | F255LANL2018 | James D Brunner |
| Los Alamos National Laboratory | Center For Nonlinear Studies | James D Brunner |
| Mayo Clinic | | Nicholas Chia |

The funders had no role in study design, data collection and interpretation, or the decision to submit the work for publication.

## Author contributions
James D Brunner, Conceptualization, Data curation, Software, Formal analysis, Investigation, Visualization, Methodology, Writing - original draft, Writing - review and editing; Nicholas Chia, Conceptualization, Resources, Formal analysis, Supervision, Funding acquisition, Investigation, Methodology, Writing - original draft, Project administration, Writing - review and editing

## Author ORCIDs
James D Brunner http://orcid.org/0000-0002-8147-2522
Nicholas Chia http://orcid.org/0000-0001-9652-691X

## Decision letter and Author response
Decision letter https://doi.org/10.7554/eLife.83690.sa1
Author response https://doi.org/10.7554/eLife.83690.sa2

---

# Additional files

## Supplementary files
• MDAR checklist

• Supplementary file 1. Supplementary tables. (**Relative_Abundance_Table**) Relative abundance of each taxa in each sample in the data-set, product of Bracken analysis on original data. (**RefSeq_Genomes_Used**) Genomes matched to taxa in data from RefSeq database, used to create models. (**Taxa_Names**) Names of taxa in the data, matched with Taxa ID. (**Baseline_Sample_Coverage**) Coverage (as proportion of relative abundance) of models used in analysis of Baseline time point samples (i.e. taxa for which we could identify a high quality close match genome). (**Treatment_Sample_Coverage**) Coverage (as proportion of relative abundance) of models used in analysis of Treatment time point samples (i.e. taxa for which we could identify a high quality close match genome). (**EU_Average_Diet**) Main media file used, from vmh.life. (**Probiotic_Cell_Counts**) Cell counts of *B. longum* probiotic, provided by Maldonado-Gomez et al. (**Paramater_Sensitivity**) Summary of parameter sensitivity results (average and variance across samples). (**Baseline_Sample_Sensitivity**) Parameter sensitivity in baseline time-point samples (2 strongest positive and 2 strongest negative edges, 2 strongest positive and 2 strongest negative edges with *B. longum* as target). (**Treatment_Sample_Sensitivity**) Parameter sensitivity in treatment time-point samples (2 strongest positive and 2 strongest negative edges, 2 strongest positive and 2 strongest negative edges with *B. longum* as target).

## Data availability
Data from the study that we used to evaluate our method can be found at the following source: https://www.ncbi.nlm.nih.gov/bioproject/PRJNA324129/ (*Maldonado-Gómez et al., 2016*). Our method is implemented as the friendlyNets package, available for download at https://github.com/lanl/friendlyNets (copy archived at *Brunner, 2024*) along with re-formatted data and python scripts for the analysis found in this paper.

The following previously published dataset was used:

| Author(s) | Year | Dataset title | Dataset URL | Database and Identifier |
|---|---|---|---|---|
| Maldonado-Gómez MX, Martínez I, Bottacini F, O'Callaghan A, Ventura M, van Sinderen D, Hillmann B, Vangay P, Knights D, Hutkins RW, Walter J | 2016 | Human fecal microbiome before and after consumption of AH1206 - Raw sequence reads | https://www.ncbi.nlm.nih.gov/bioproject/PRJNA324129 | NCBI BioProject, PRJNA324129 |

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
