## [Editor Report]

This manuscript uses genome-scale metabolic modeling to estimate interspecies interactions and subsequently assess engraftment outcomes. This is an important line of work with potentially broad applications in different fields, including microbiota studies. The authors provide solid evidence to support the usefulness of their proposed approach in engraftment studies.

---

## [Decision Letter]

**Decision letter after peer review:**

Thank you for submitting your article "Metabolic Model-based Ecological Modeling for Probiotic Design" for consideration by *eLife*. Your article has been reviewed by 3 peer reviewers, including Babak Momeni as Reviewing Editor and Reviewer #1, and the evaluation has been overseen by a Reviewing Editor and Aleksandra Walczak as the Senior Editor. The following individual involved in the review of your submission has agreed to reveal their identity: Daniel Rios Garza (Reviewer #2).

Essential revisions:

1) Please include the limitations of GSMMs, such as the requirement for high-quality reconstruction and the need to have resolved genomes or a method to assign models to taxa (Figure 1). Please also the assumption of the defined medium used for this manuscript.

2) Please add an investigation of the impact of the growth media on the predictions to the manuscript.

3) Please clarify the details of the setup for the results in Figures 3, 4, and 5. How is the training set chosen? How is the test set chosen? For testing each model, what are the conditions explored? What is the justification behind these choices?

4) Please clarify the importance of Figure 3. It appears all models, even the simplest ones, accurately predict the outcomes in this case. What is unique/different about this example that leads to this situation?

5) Please add explanations for why the performance in Figures 4 and 5 is rather poor. What is the reason behind the low predictive power? There is also a need for explaining why some models perform better than others. What are the conditions under which you expect each model to do well?

6) Please include the detailed assumptions of your FBA model, in particular, the effect of the resource allocation threshold. If a fixed threshold is used for research allocation, it may lead to artifacts. Please specify how this threshold affects the AUCs and if a more general approach such as parsimonious FBA would give similar results.

7) A justification is needed for why log ratios are used to represent interaction coefficients (Equation 2). When does such a representation work well? Does this work well, for example, in a simple case of a community with a few interacting species in dynamic FBA?

8) The authors need to justify how they chose the species to include in the analysis ("Creating induced sub-graphs"). If the availability of GSMMs is the only criterion, the authors need to add a thorough investigation of how sub-sampling the species present in the community might impact the prediction quality.

9) Please clarify how exactly the 400 cases used in each case are drawn (by clarifying and expanding the description in "approximate edge swapping").

10) Please revise the Discussion section to address the limitations of the approach.

Additionally, I strongly encourage you to address the comments from individual reviewers (listed in the following) in your revision.

*Reviewer #1 (Recommendations for the authors):*

The authors in this manuscript use different models to explore if engraftment success can be predicted based on metabolic interactions. They use genome-scale metabolic models to train six different population-level models. They then assess how well these models predict the invasion/engraftment outcomes when compared with existing experimental results. The basic concept behind the work is interesting, relevant, and timely. However, there are issues with the organization of the paper and the clarity of results which make the paper hard to evaluate. Additionally, there are several assumptions made in the paper (e.g. how the interactions are estimated or how instances of communities are built) that need to be better justified. A thorough discussion about the limitations of the approach is also necessary to clarify for the reader the situations in which this approach is likely (or unlikely) to work well.

1. It is unclear to me what the ensemble used for Figures 3 to 5 is. How is the training set chosen? How is the test set chosen? For testing each model, what are the conditions explored?

2. Is it fair to consider the results in Figure 3 as trivial? It appears all models, even the simplest ones, accurately predict the outcome. What is unique/different about this example that leads to this situation?

3. In contrast to Figure 3, the results in Figures 4 and 5 are far from ideal. What is causing this significant drop in predictive power? I would recommend that the authors include the discussions to at least speculate under what conditions the predictive power will drop.

4. In the organization of the paper, there is a strong emphasis on the six different types of model in the abstract and introduction, but the most informative data comes from the InhibitLV model. I felt the models that are not informative could perhaps be de-emphasized, in favor of models that produce the best predictions and offer more mechanistic investigations.

5. In my opinion, more details about how FBA is used to find the parameters of the model should be included in the main text.

6. The authors briefly mention that in some cases one or more models failed because of fitting issues. In my opinion, including the conditions that each model fails as well as a description of why it fails and how the situation can be remedied is necessary. This will be very informative for others to troubleshoot and fix similar issues in other contexts.

7. If I understand correctly, the authors use log ratios to represent interaction coefficients (Equation 2). In my opinion, the justification for this choice needs to be included in the manuscript. Does this work well, for example, in a simple case of a community with a few interacting species in dynamic FBA?

8. It appears to me that in choosing what species to include in the analysis ("Creating induced sub-graphs"), species to include are chosen based on whether there is a match for them in the AGORA database, rather than how important they are for the process of engraftment. The potential downside of this choice for accurately representing each system can be large (for example, it may miss the dominant species as an obvious example). The authors need to include a justification of why this choice is justified.

9. I would like to ask for some clarification on how exactly the 400 cases used in each case are drawn. The procedure is described in lines 423-432 ("approximate edge swapping"), but I think expanding the description (maybe using a simplified visual illustration of the protocol with 4 or 5 species) would be helpful.

10. The Discussion section of the paper summarizes the advantages of the proposed method (using genome-scale metabolic modeling to make inferences about engraftment outcomes) but does not adequately discuss the limitations of the approach. In my opinion, this is one of the major weaknesses of the paper.

*Reviewer #2 (Recommendations for the authors):*

The study presents a promising framework to predict the outcome of colonization experiments, such as the ones that are attempted in probiotic therapy. Previous studies have tackled this problem with both machine learning approaches (ML) and ecological models (based on interspecies interactions) but encountered some major limitations. The ML approaches are context-specific and require a large amount of training data per condition to have predictive power, while the ecologic models require dense time series to reliably find the interspecies interaction parameters.

The authors propose to solve these limitations by making use of reconstructed genome-scale metabolic models (GSMMs) to predict the strength and direction of species interactions. The main advantage of such an approach is the possibility to fit ecological models to individual samples, without the need for extensive training data or dense time series. But does this approach work?

To test its strength, the authors predicted the outcome of three colonization experiments and show that the ecological models parameterized with GSMMs outperform ML methods such as random forest and support vector machines.

While the approach is certainly promising, some careful validations might still be useful before it can be widely used.

Even if the framework seemingly outperforms the tested ML methods, the predictive values are relatively low and might depend on details of the approach that were not systematically evaluated. High-quality GSMMs are not widely available. Automated and semi-automated reconstructions such as the ones in AGORA contain many errors and might not provide the correct information about species growth and interactions. The study partially tackles this limitation by showing that predictions using the interactions of a null model (i.e. random interactions) are significantly worse than predictions based on the GSMM interactions.

Of notice, the framework uses a modified version of community FBA which is not accessible in a publicly available tool and relies on the arbitrary definition of a flux threshold. Furthermore, the use of GSMMs requires the formulation of the growth media. Here an artificial formulation (termed the Western diet) was used. It is unclear how much these choices affect the prediction quality and prevent the framework from being widely used.

How about the ecological models? The authors extensively tested six alternatives divided into two classes: generalized Lotka-Volterra-based models (GLVs) and linear models. Overall, the GLVs outperformed the linear models. Finally, if the approach can be used and makes reliable predictions, then one can perform perturbations to the structure of the model and gain further insights into the ecology of a species' colonization into a community.

In my opinion, the following additional validations would significantly increase the impact of the study and strengthen its conclusions:

– Growth media: the manuscript would benefit by showing how much the growth media affects the quality of the predictions;

– Resource allocation threshold: It's not specified in the methods, but based on the previous study, I assume that a threshold was fixed, which sounds fairly artificial. It would be useful to know how this threshold affects the AUCs and if a more general approach such as parsimonious FBA would give similar results.

– Induced sub-graphs: what is the consequence of working with only a fraction of the microbiome given the impossibility to map it to a GSMM? For instance, how does down-sampling the community composition impact the prediction quality?

With these comparisons in hand, one could understand better why the AUCs are so low and what is expected to get reliable predictions.

Figure 1: could make clear some of the limitations of GSMMs, such as the requirement of high-quality reconstruction and the need to have resolved genomes or a method to assign models to taxa. Also, the assumption of a defined medium.

Figure 3: why is the null model not centered at 0.5?

*Reviewer #3 (Recommendations for the authors):*

Recent years have shown that despite the strong link between the microbiome and health, the administration of a probiotic is not sufficient to alter the microbiome in a desired, therapeutic, way. Engraftment of the administered microbial species is often limited, with most patients failing to show signs of medium-term or long-term engraftment. This study uses pairwise metabolic modeling to build a network of species-species interactions. Considering three experimental engraftment studies, they assess the likelihood of invader engraftment based on network structure, comparing model predictions to data. Comparing several related models, it is shown that a generalized Lotka-Volterra model is predictive if an invader or probiotic will successfully engraft into an existing microbiota community, potentially revealing which microbe-microbe interactions drive the engraftment. A known weakness of generalized Lotka-Volterra models is the difficulty to parametrize them from experimental data, and the lack of generalizability of the resulting model to new situations. Here, by leveraging metabolic modeling (relying on the AGORA network) the authors claim to parameterize population models in a way that is predictive also of novel environments. Thus, the authors claim to predict the engraftment of an invader into a microbial community.

Strengths:

– The question of engraftment is important, and the attempt to tackle it by trying several related models and AGORA is justified and may yet prove fruitful.

– Figure 2 explained the six models well, clarifying the differences and similarities between them.

– The code shared on github is good practice. Yet one may improve the code release significantly, e.g.: a readme file, a quick explanation of how to use the code, etc. In its current form, it is not so usable by a third party.

Weaknesses:

– The message in Figure 1 is lost on me. The two panels appear quite similar, and I do not understand what the difference between them is supposed to mean.

– I do not agree that the ecological principle of engraftment is the same between pathogens and probiotics. The pathogen cares only about short-term engraftment as a means to grow and infect further hosts (hence its pathogenic nature, ultimately damaging the host), therefore, the notion of long-term engraftment is ill-defined. Indeed, pathogens may employ strategies to assist in short-term engraftment (e.g. cause the host diarrhea), that probiotics cannot. Probiotics (or commensal species in general) are presumed to form stable long-term associations with the host without damaging the host.

– I found the exposition of the methods and results lacking. For a start: how precisely is AGORA used? Why do the authors presume the readers are familiar with AGORA? Equation 1 is not properly explained. The Results section shows mostly trivial information instead of a deeper understanding of the modeling results.

– It is not at all obvious that given the FBA data for a single microbe, one can predict its growth paired with another species. Processes of adaptation and evolution kick in and can change the single-microbe growth profiles significantly. This weakens the foundation of this entire approach. There is much recent literature on this co-culture question (some of it cited in the manuscript) and it simply cannot be brushed off so casually.

Justification of claims and conclusions:

The manuscript raises an important question and proposes a solid direction, but I am not convinced by the evidence presented-possibly due to the presentation itself. Though the approach proposed in this manuscript may yet prove useful, it needs much more justification, proper exposition, and experimental validation to be taken truly seriously.

– Abstract (line 20) "reflect connect" should probably be just "connect".

– While I think there is no need to explain the AUROC, I do think that the Kendall-tau correlation should be better motivated, as it is not often used in the microbiome literature.

– The AGORA database should be better exposed. The authors should not assume the reader is familiar with it.

– I do not understand why it is useful to show Figure 3a – a bunch of 100% AUROC values (as a bar plot!). The fact that it is easy to predict C.diff. engraftment is stated in the text and the figure frankly just looks a bit silly. Indeed, Figure 3b is even less informative, we know what a 100% AUROC looks like, and it needs no visual assistance. Though Figure 3c contains something for us to learn, it is a weak statement about the role of shuffling the AGORA parameters. Altogether – I would say remove 3a and 3b altogether and change 3c or put it in a supplement.

– Regarding Figure 4a: the use of bar plots is discouraged, using box plots would be a lot more informative in the same presentation style and space. 4b: why is the AUROC not stated? The classifier seems hardly impressive.

---

## [Author Response]

Essential revisions:1) Please include the limitations of GSMMs, such as the requirement for high-quality reconstruction and the need to have resolved genomes or a method to assign models to taxa (Figure 1). Please also the assumption of the defined medium used for this manuscript.

We have added the following regarding the limitations of GSMMs in the introduction (lines 77-88):

“Genome-scale metabolic models (GSMs) encode the metabolic pathways available in a cell, allowing simulation of cellular metabolism and growth. These tools have been used extensively to understand and engineer microbial mono-cultures, and have recently begun to be used to understand and predict the composition and metabolic function of microbial communities. Community methods using GSMs often suffer from a number of drawbacks. These can include a focus on equilibrium states, high complexity and computational cost, and the difficulty of providing accurate GSMs for each community member. By layering simple, pairwise GSM community modeling with population models, we allow for prediction that includes transient behavior (e.g. how quickly an extinction happens within a community) and reduces computational complexity. The availability of accurate GSMs remains a major difficulty, but we demonstrate that progress can be made using algorithmically generated models from whole genomes”

And in the discussion (lines 242-249):

“As a limitation, it is worth noting that all GEM-based methods, including our own, are limited to a large extent by the accuracy of the metabolic model reconstructions. Low-quality reconstructions often omit key reactions, or worse, may include reactions that should not have been included. In all cases, this creates the potential for reaction fluxes or interactions that do not reflect reality. Improving the quality of metabolic model reconstruction is the subject of challenging and nuanced research that includes automated pipelines and hand-curation efforts. These efforts are essential to improve the accuracy of community metabolic modeling, but are beyond the scope of this particular study.”

We note that this work makes no attempt to correct the inherent limitations and that our results are positive despite these many known limitations. In our opinion, this increases the significance of this work.

We have also added the following detail about how we now reconstruct models (note that this method is different from our previous practice of simply matching taxa to models in the AGORA database) in the Introduction (lines 92-97):

“To construct the necessary GSMs, we matched taxa found in the sample to the genomes found in the RefSeq database and used these genomes to build models using ModelSEED. A complete list of the genomes for each data-set used can be found in Supplementary file S1 Data.”

As well as further details in the Methods section, in the section labeled “Creation of GSMs”.

The defined medium is also now listed in the supplemental data file S1, sheet labeled “EU_Average_Diet”

2) Please add an investigation of the impact of the growth media on the predictions to the manuscript.

Our original results used a gut media defined by the Theile group and is available as part of the Virtual Metabolic Human database. We highlight that media components were not hand-picked by the authors and there is no risk of “overfitting” by hand as it were. Given we cannot test everything, we offer to reduce this question to two main questions that a reader might have.

1) Does the model respond appropriate to change in media?

2) Is the model overly sensitive to small changes in media?

For the first question, we have replaced the gut media with complete media. This essentially is akin of changing from a media meant to mimic a (micro)aerobic colon environment to an aerobic nutrient-rich laboratory culture.

For the second question, we have halved and doubled the flux rates (the typical stand in for concentration) of the major carbon and nitrogen sources from the gut media and report the difference.

We have added both of these results to manuscript in table 1 and figure 2.

3) Please clarify the details of the setup for the results in Figures 3, 4, and 5. How is the training set chosen? How is the test set chosen? For testing each model, what are the conditions explored? What is the justification behind these choices?

We thank the reviewer for these remarks. We have clarified the text. Specifically, we have highlighted that our model is not a “learned” model where there is a training set and a test set, but instead these are models whose parameters come entirely from a database of metabolic models. In fact, the lack of fitting parameters is one of the wonderful things about this model in that makes it potentially more extensible and able to directly incorporate genomic and literature knowledge in the form of genome scale metabolic models! We have added the following to our methods section entitled “Prediction and evaluation” (lines 399-402)

“We note that our method ``learns" using joint FBA, and so there is no need to split the data into training and testing sets. In fact, the ``learning" procedure does not make use of any known sample classifications, as a traditional machine-learning algorithm would. This allows us to test our predictions on the entire data set, which provides greater confidence in our results.”

The comparison models, that is, the SVM and RF models, on the other hand, are learned models. We had added the following details for those models (lines 426-433):

“We compared the performance of our method to the performance of two traditional machine learning approaches, support vector machines and random forest classifiers. We used $1000$-fold cross validation of the predictions made by these tools (i.e. $1000$ independent train/test splits). Each trial consisted of splitting the data randomly into 16 training samples and 6 testing samples, fitting both a support vector machine and a random forest classifier to the training samples using standard python libraries (from scikit-learn v. 0.23.2), and evaluating the model's predictions of the testing samples. This was repeated for both the ``baseline" and ``treatment" sets of samples.”

Our aim was to use these as a potential estimate of a reasonable upper limit for our model which has far fewer free parameters and which does not learn from data. As is shown in our figures, our approach is comparable or slightly exceeds that of these standard fitting approaches.

4) Please clarify the importance of Figure 3. It appears all models, even the simplest ones, accurately predict the outcomes in this case. What is unique/different about this example that leads to this situation?

We have removed Figure 3 and the corresponding text from the manuscript. This example, while it performs well, could indeed be thought of as a trivial predictor given the easily separable microbiome distributions between susceptible and non-susceptible microbiomes. The original paper shows these are easily visually separated on a PCoA with almost no overlap and little variance per group and is in Author response image 1.

**Author response image 1. sa2fig1:** 

Instead, we now focus on the most complex case in order to provide a better sense of the performance on a case where there is a more reasonable need for an improved mechanistic predictor. This case is one where there is no clean separation. Pasting the figure from the author’s original paper in Author response 2.

5) Please add explanations for why the performance in Figures 4 and 5 is rather poor. What is the reason behind the low predictive power? There is also a need for explaining why some models perform better than others. What are the conditions under which you expect each model to do well?

We refer to reviewers to our answer to the previous question for which we added a paragraph discussing the differences between the different experimental scenarios. We also now focus on the results from Figure 4, which allowed us to have the highest model coverage and test the most complex case. (And removed the discussion of Figure 5 in part because of the amount of additional work that was already performed and the extensive amount of time it therefore took us to respond to the reviewer requests regarding the methodology.)

6) Please include the detailed assumptions of your FBA model, in particular, the effect of the resource allocation threshold. If a fixed threshold is used for research allocation, it may lead to artifacts. Please specify how this threshold affects the AUCs and if a more general approach such as parsimonious FBA would give similar results.

We thank the reviewers for this astute comment. We previously performed these experiments and did not include them in the interest of clarity and succinctness. But we are quite happy to bring this work to light as we believe this highlights the strength of our results. We have now provided results using a different RAC or no RAC at all the manuscript (table 1, figure 2). The results show that RACs do not improve performance significantly, although we note that RAC has been shown to improve prediction in pairwise experiments in previous studies.

We would like to briefly note the following as part of our results and rationale as strengths. As we indicate in our paper on RACs, a fixed threshold is most likely not biologically true. However, we lack the data to validate species-specific RACs which might be more biologically appropriate. By purposefully simplifying our model of RACs to a single fixed threshold, we are avoiding the possibility that these essentially free parameters could be used to fit our data and inflate the goodness of our results. Another way of thinking about this is that we are purposely giving up potential performance in order to convince the reader that the positive results stem from the metabolic models and RACs, and not due to a significantly larger number of extraneous parameters. With more data and more precise parameterization of the RACs, could we do better? Almost definitely. But our purpose is to highlight the power of the models with minimum free parameter fitting. Arguably, the fixed threshold chosen is also not a free parameter as it is chosen on the basis of maximizing overall diversity as discussed in the RAC paper.

7) A justification is needed for why log ratios are used to represent interaction coefficients (Equation 2). When does such a representation work well? Does this work well, for example, in a simple case of a community with a few interacting species in dynamic FBA?

We now include the following explanation in our methods section (lines 363-372):

“We use the log-ratio of simulated growth in pairs and alone as the implied effect of one microbe on another. That is, if simulated growth of species *i* alone is *x_i_* 364 , and growth of species *i* when coupled in a pair with species *j* is *x_ij_*, we weight the edge from species *j* to species *i* as

wji∗=log⁡(xijxi) We use the log-ratio because it is the simplest choice of translating simulated data to interaction that accounts for fold-changes. Of course, the simple difference in simulated biomass is another possible choice, but this will be much more sensitive to absolute scaling in simulation, so, for example, slow growing taxa will be biased towards smaller interactions in the network. Furthermore, our *in-silico* growth experiments only provide simulated biomass values at chemostatic equilibrium, meaning that fitting Lotka-Volterra parameters directly to these data is an underdetermined problem, particularly when pair growth leads to a simulated extinction.”

We note that dynamic FBA implies a more complicated relationship between taxa that is mediated by available metabolites.

8) The authors need to justify how they chose the species to include in the analysis ("Creating induced sub-graphs"). If the availability of GSMMs is the only criterion, the authors need to add a thorough investigation of how sub-sampling the species present in the community might impact the prediction quality.

This is a good point. Our previous rationale was based on availability of hand-curated AGORA models that were exact matches, and therefore, in part, was limited by accessibility. In our revisions, we have now added ModelSEED reconstructed models, which means we carried out the steps of draft model reconstruction and gap filling on approximately 1000 additional models.

To the methods, we have added (lines 381-388):

“In order to carry out our analysis, we needed to construct GSMs for the taxa found in the data. To do this, we matched taxa in the data by NCBI taxa ID to genomes in the NCBI refseq database (O’Leary et al., 2016). We matched taxa to the nearest (by tree distance according to NCBI's taxonomic database) genome that was listed as ‘Chromosome’ or ‘Complete Genome’, or contained at least 1000 genes. On average, this approach allowed us to perform our experiments with ~ 87-88% coverage of each sample. Taxa for which we failed to match a genome were removed from the data, and all analysis was performed on the remaining. For full details on the coverage of each sample, see Supplementary file S1 Data.”

Our new methodology gives us a higher relative abundance representation of the overall community, taking us to 87-88% coverage per sample in our revised manuscript. This new methodology does not give substantially better performance, but does provide much more complete model coverage.

9) Please clarify how exactly the 400 cases used in each case are drawn (by clarifying and expanding the description in "approximate edge swapping").

This has been clarified in the methods section with the following text (lines 414-424):

“We estimated the significance of our predictions against predictions from a null-model created by parameterizing a Lokta-Volterra system from an ‘edge swapped’ full network. The ‘edge-swapped’ network was constructed by drawing edge weights from the original full network, with replacement, as is commonly done to create a null model in network analysis (Röttjers et al., 2021). This approach allows us to create random models that preserve the statistical characteristics of the models that we test on, while changing the specific interactions between taxa. That is, the distribution of the entire set of parameters in the null models βijN will be approximately the same as the distribution of parameters βij determined by joint FBA. Using this null model ensures that our conclusions are drawn from the joint FBA parameter learning method, and not simply a result simpler characteristics, e.g. total connectedness of the community.”

10) Please revise the Discussion section to address the limitations of the approach.

In response to points 1 and 4 above, we have added text regarding the limitations of GSMMs in general as well as a discussion of limitations regarding microbial species representation in the database (which are generally intended for human microbiota).

Reviewer #1 (Recommendations for the authors):The authors in this manuscript use different models to explore if engraftment success can be predicted based on metabolic interactions. They use genome-scale metabolic models to train six different population-level models. They then assess how well these models predict the invasion/engraftment outcomes when compared with existing experimental results. The basic concept behind the work is interesting, relevant, and timely. However, there are issues with the organization of the paper and the clarity of results which make the paper hard to evaluate. Additionally, there are several assumptions made in the paper (e.g. how the interactions are estimated or how instances of communities are built) that need to be better justified. A thorough discussion about the limitations of the approach is also necessary to clarify for the reader the situations in which this approach is likely (or unlikely) to work well.

We thank Reviewer #1 for these comments, especially for the many pointers to where we can better clarify or expand upon our results to the benefit of the readers.

1. It is unclear to me what the ensemble used for Figures 3 to 5 is. How is the training set chosen? How is the test set chosen? For testing each model, what are the conditions explored?

Thank you. Addressed above in Essential revisions #3.

2. Is it fair to consider the results in Figure 3 as trivial? It appears all models, even the simplest ones, accurately predict the outcome. What is unique/different about this example that leads to this situation?

Thank you. Addressed above in Essential revisions #4.

3. In contrast to Figure 3, the results in Figures 4 and 5 are far from ideal. What is causing this significant drop in predictive power? I would recommend that the authors include the discussions to at least speculate under what conditions the predictive power will drop.

Thank you. Addressed above in Essential revisions #4.

4. In the organization of the paper, there is a strong emphasis on the six different types of model in the abstract and introduction, but the most informative data comes from the InhibitLV model. I felt the models that are not informative could perhaps be de-emphasized, in favor of models that produce the best predictions and offer more mechanistic investigations.

We now de-emphasize the 6 different types of models and note for the reviewer that using the new method of finding models, we no longer have this sensitivity. It is perhaps worth noting that though all the LV models are now roughly equivalent, InhibitLV was faster than the others because it made finite time blow-up less likely. Perhaps another way of putting this is to say the reviewer suggestion to push for higher coverage, though it required fairly large amount of work, improved the model quality, especially in terms of sensitivity (and therefore also adding to the confidence we have in the results).

5. In my opinion, more details about how FBA is used to find the parameters of the model should be included in the main text.

Thank you. We note that the methods used to find parameters are not novel, and were previously introduced in the RAC paper by Minsuk Kim et al. That said, we have now included the following brief recap so that the reader is provided high-level insight on the methodology from our previous work (lines 334-344).

“To compute growth simulations, we use joint FBA with a set of resource allocation constraints, which have been shown to improve predictions in pairwise experiments (Kim et al., 2022). Briefly, joint FBA with resource allocation constraints solves a linear program defined by the GSMs of the community members. The linear program maximizes total community biomass with the standard FBA constraints as well as the added constraints that each microbe grows a fixed known rate (simulating chemostatic equilibrium), that mass is balanced community wide (with inflow and outflow of nutrients), that there is forced leak of nutrients out of the community (introduced by Kim et al., 2022 to prevent unrealistically efficient cross-feeding) and that each community member’s total internal reaction flux is bounded (the resource-allocation constraint). Our problem set-up is identical to that described in Kim et al., 2022, and we provide python code to carry out the computation in our supplemental repository.”

6. The authors briefly mention that in some cases one or more models failed because of fitting issues. In my opinion, including the conditions that each model fails as well as a description of why it fails and how the situation can be remedied is necessary. This will be very informative for others to troubleshoot and fix similar issues in other contexts.

Our new method of identifying and including models in our analysis provided much better sample coverage, alleviating many of these issues. One issue that remains is the problem of finite-time blow-up in simulation. We now including the following discussion of how the model might fail due to finite-time blow-up (lines 163-175):

“One consequence of using the generalized Lotka-Volterra model in our method is the possibility of finite-time blow-up in simulation. This happens when one or more taxa approach infinite biomass in finite simulation time. In practice, whenever one or more taxa increases to a level much higher than the initial total biomass of the simulated system, simulation is slowed down and results may be less reliable. There are two simple methods to prevent this from happening. The first is to alter every interaction parameter to be more negative (see Equation 3), making the system uniformly more antagonistic and less likely to show finite-time blow-up. The second is to include negative self-inhibition (e.g. add βii<0 parameters [see Equation 4]).

We tested both of these alterations by implementing them and compuing predictions and the AUC-ROC of those predictions. \Cref{fig:lvshift} shows that these changes did not have a significant impact on the performance of our predictions, as long as only antagonism was considered. When we shifted parameters in the positive direction (simulating commensalism or mutualism), finite-time blow-up was more likely and our results were much less stable.”

7. If I understand correctly, the authors use log ratios to represent interaction coefficients (Equation 2). In my opinion, the justification for this choice needs to be included in the manuscript. Does this work well, for example, in a simple case of a community with a few interacting species in dynamic FBA?

Thank you. Addressed in Essential revisions #7.

8. It appears to me that in choosing what species to include in the analysis ("Creating induced sub-graphs"), species to include are chosen based on whether there is a match for them in the AGORA database, rather than how important they are for the process of engraftment. The potential downside of this choice for accurately representing each system can be large (for example, it may miss the dominant species as an obvious example). The authors need to include a justification of why this choice is justified.

Thank you. Addressed in Essential revisions #8.

9. I would like to ask for some clarification on how exactly the 400 cases used in each case are drawn. The procedure is described in lines 423-432 ("approximate edge swapping"), but I think expanding the description (maybe using a simplified visual illustration of the protocol with 4 or 5 species) would be helpful.

Thank you. Addressed in Essential revisions #9.

10. The Discussion section of the paper summarizes the advantages of the proposed method (using genome-scale metabolic modeling to make inferences about engraftment outcomes) but does not adequately discuss the limitations of the approach. In my opinion, this is one of the major weaknesses of the paper.

Thank you. Addressed in Essential revisions #10. The senior author would like to note that any perceived lessening of the limitations of metabolic modeling was unintentional. The senior author has a reasonably long history working on tools for or related to automated metabolic modeling and it erroneously stopped occurring to him that the limitations and difficulties with metabolic models would be unknown by general readers. On the flip side, the excitement of this working as it does despite those many limitations is potentially very high.

Reviewer #2 (Recommendations for the authors):The study presents a promising framework to predict the outcome of colonization experiments, such as the ones that are attempted in probiotic therapy. Previous studies have tackled this problem with both machine learning approaches (ML) and ecological models (based on interspecies interactions) but encountered some major limitations. The ML approaches are context-specific and require a large amount of training data per condition to have predictive power, while the ecologic models require dense time series to reliably find the interspecies interaction parameters.The authors propose to solve these limitations by making use of reconstructed genome-scale metabolic models (GSMMs) to predict the strength and direction of species interactions. The main advantage of such an approach is the possibility to fit ecological models to individual samples, without the need for extensive training data or dense time series. But does this approach work?To test its strength, the authors predicted the outcome of three colonization experiments and show that the ecological models parameterized with GSMMs outperform ML methods such as random forest and support vector machines.While the approach is certainly promising, some careful validations might still be useful before it can be widely used.Even if the framework seemingly outperforms the tested ML methods, the predictive values are relatively low and might depend on details of the approach that were not systematically evaluated. High-quality GSMMs are not widely available. Automated and semi-automated reconstructions such as the ones in AGORA contain many errors and might not provide the correct information about species growth and interactions. The study partially tackles this limitation by showing that predictions using the interactions of a null model (i.e. random interactions) are significantly worse than predictions based on the GSMM interactions.Of notice, the framework uses a modified version of community FBA which is not accessible in a publicly available tool and relies on the arbitrary definition of a flux threshold. Furthermore, the use of GSMMs requires the formulation of the growth media. Here an artificial formulation (termed the Western diet) was used. It is unclear how much these choices affect the prediction quality and prevent the framework from being widely used.How about the ecological models? The authors extensively tested six alternatives divided into two classes: generalized Lotka-Volterra-based models (GLVs) and linear models. Overall, the GLVs outperformed the linear models. Finally, if the approach can be used and makes reliable predictions, then one can perform perturbations to the structure of the model and gain further insights into the ecology of a species' colonization into a community.

We thank Reviewer #2 for this insightful summary of our work as well as the questions raised. We appreciate these remarks and agree with the overall spirit of them. We briefly address the highlights here. First, the authors agree (without throwing any stones at VMH—whose services we appreciate) that there are many errors in the AGORA models. That being said, a metabolic model can be considered at multiple scales, that is, more reductively on the reaction level or more holistically on the growth level. On the reaction level, one can find many errors of both inclusion and exclusion within AGORA models. For example, the specific electron transport chain reactions might be erroneous. However, on the growth level, anaerobes can grow without oxygen, and aerobes use oxygen, and overall the gross growth characteristics have been mostly preserved. We note this not to agree or disagree with the reviewer on the overall quality of the models, but to point out that our model is at the population level and therefore not very sensitive to the most erroneous aspect of the AGORA models, which occur at the reaction level. Another way of saying this is that most errors in reaction pathways will not have a strong effect on species-species interactions unless they happen to impact a key metabolite.

We also note, though it is not described in this paper, that the flux threshold for the RAC is not arbitrary, but chosen on the basis of maximizing diversity as discussed in the paper by Minsuk Kim et al. This is something that needed some clarity and we have now provided a brief summary of that work in this paper as addressed in Essential revision #6. We acknowledge that we used a variation of SteadyCom where we added a RAC to calculate the species-species interaction parameters and that there is no specific software package. But this is mostly because the change is very minor and are essentially not worthy of its own software implementation. In lieu of this, we now include this algorithm in our software package containing this method.

The media file is indeed a good question and one that we address in Essential revision #2. And finally, the excitement of the mechanistic model is that indeed, we can perturb the model and identify the species that most alter engraftment and non-engraftment. This is what we show an example of in Tables 1 and 2. This is a powerful way to generate hypotheses and while the aim of our work was not to explore the biological questions, it is one of the strengths of this modeling paper and a goal for future papers.

In my opinion, the following additional validations would significantly increase the impact of the study and strengthen its conclusions:– Growth media: the manuscript would benefit by showing how much the growth media affects the quality of the predictions;

Thank you. Addressed in Essential revisions #5.

– Resource allocation threshold: It's not specified in the methods, but based on the previous study, I assume that a threshold was fixed, which sounds fairly artificial. It would be useful to know how this threshold affects the AUCs and if a more general approach such as parsimonious FBA would give similar results.

Thank you. Addressed in Essential revisions #6.

– Induced sub-graphs: what is the consequence of working with only a fraction of the microbiome given the impossibility to map it to a GSMM? For instance, how does downsampling the community composition impact the prediction quality?

Thank you. Addressed in Essential revisions #8.

With these comparisons in hand, one could understand better why the AUCs are so low and what is expected to get reliable predictions.Figure 1: could make clear some of the limitations of GSMMs, such as the requirement of high-quality reconstruction and the need to have resolved genomes or a method to assign models to taxa. Also, the assumption of a defined medium.

We have now made clear the limitations of GSMMs in the Introduction and Discussion in response to Essential revision #1. We had added to the discussion of the defined media in our response to Essential revision #2.

Figure 3: why is the null model not centered at 0.5?

This is because the null model is essentially a bootstrap in the sense that we take the parameters of the real model and shuffle them around. The edge weights are therefore not random and most microbes suppress *C. difficile*. We now remove Figure 3 and the discussion of this simpler case entirely in response to other remarks.

Reviewer #3 (Recommendations for the authors):Recent years have shown that despite the strong link between the microbiome and health, the administration of a probiotic is not sufficient to alter the microbiome in a desired, therapeutic, way. Engraftment of the administered microbial species is often limited, with most patients failing to show signs of medium-term or long-term engraftment. This study uses pairwise metabolic modeling to build a network of species-species interactions. Considering three experimental engraftment studies, they assess the likelihood of invader engraftment based on network structure, comparing model predictions to data. Comparing several related models, it is shown that a generalized Lotka-Volterra model is predictive if an invader or probiotic will successfully engraft into an existing microbiota community, potentially revealing which microbe-microbe interactions drive the engraftment. A known weakness of generalized Lotka-Volterra models is the difficulty to parametrize them from experimental data, and the lack of generalizability of the resulting model to new situations. Here, by leveraging metabolic modeling (relying on the AGORA network) the authors claim to parameterize population models in a way that is predictive also of novel environments. Thus, the authors claim to predict the engraftment of an invader into a microbial community.Strengths:– The question of engraftment is important, and the attempt to tackle it by trying several related models and AGORA is justified and may yet prove fruitful.

We thank the reviewer for this comment.

– Figure 2 explained the six models well, clarifying the differences and similarities between them.– The code shared on github is good practice. Yet one may improve the code release significantly, e.g.: a readme file, a quick explanation of how to use the code, etc. In its current form, it is not so usable by a third party.

We have improved the github by adding detailed documentation. It is more of a calculation than software, but hopefully in adding an example we will provide enough information for any future users who may want to do their own calculations.

Weaknesses:– The message in Figure 1 is lost on me. The two panels appear quite similar, and I do not understand what the difference between them is supposed to mean.

We have edited the caption for clarity. Our purpose with this figure was to immediately draw attention to the mechanistic nature of the GSM approach, in contrast to standard parameter learning from data.

– I do not agree that the ecological principle of engraftment is the same between pathogens and probiotics. The pathogen cares only about short-term engraftment as a means to grow and infect further hosts (hence its pathogenic nature, ultimately damaging the host), therefore, the notion of long-term engraftment is ill-defined. Indeed, pathogens may employ strategies to assist in short-term engraftment (e.g. cause the host diarrhea), that probiotics cannot. Probiotics (or commensal species in general) are presumed to form stable long-term associations with the host without damaging the host.

This is a very astute point as there are indeed differences between pathogens and probiotics, for example, the role of the immune response and the effect on the motility of the host differs between them in magnitude. We included it initially because there are a limited number of probiotic experiments with corresponding microbiome data publicly available. We have removed the discussion of the *C. difficile* case.

– I found the exposition of the methods and results lacking. For a start: how precisely is AGORA used? Why do the authors presume the readers are familiar with AGORA? Equation 1 is not properly explained. The Results section shows mostly trivial information instead of a deeper understanding of the modeling results.

We thank the reviewer for these remarks. While we did do a lot of work to ensure the quality of our approach, we clearly tried to over-simplify the paper in the interest of clarity and brevity. We have added a lot of our results regarding model variants and parameter modification into the supplemental so that the reader can appropriately understand the underlying models. In addition, we have tried to add additional background in-line where possible to try to limit the reliance on the readers having familiarity with the citations.

– It is not at all obvious that given the FBA data for a single microbe, one can predict its growth paired with another species. Processes of adaptation and evolution kick in and can change the single-microbe growth profiles significantly. This weakens the foundation of this entire approach. There is much recent literature on this co-culture question (some of it cited in the manuscript) and it simply cannot be brushed off so casually.

We thank the reviewer for this astute comment. It was not our intent to brush off the complexity of species-species interactions which, as the reviewer states, are highly nonlinear. It is indeed surprising that despite all the caveats we have acknowledged in our work that this approach yields positive predictive power. It is for this reason, we feel our results are worth highlighting. That is, we are not of the position that this model is the be-all end-all of engraftment studies, but that it is surprisingly effective and this yields both biological insight into the importance of metabolism, the ability of approximate metabolic models to reflect the right properties at a community level, and that we do not need models with large numbers of free parameterized models to capture this information.

Justification of claims and conclusions:The manuscript raises an important question and proposes a solid direction, but I am not convinced by the evidence presented-possibly due to the presentation itself. Though the approach proposed in this manuscript may yet prove useful, it needs much more justification, proper exposition, and experimental validation to be taken truly seriously.

We thank the reviewer for this comment and the other reviewers as well who asked for us for this justification and exposition, which we have now hopefully provided. Experimental validation is unfortunately outside the scope of this theoretical work, but we do note that we used 3 datasets and a model with no free parameters.

– Abstract (line 20) "reflect connect" should probably be just "connect".– While I think there is no need to explain the AUROC, I do think that the Kendall-tau correlation should be better motivated, as it is not often used in the microbiome literature.– The AGORA database should be better exposed. The authors should not assume the reader is familiar with it.– I do not understand why it is useful to show Figure 3a – a bunch of 100% AUROC values (as a bar plot!). The fact that it is easy to predict C.diff. engraftment is stated in the text and the figure frankly just looks a bit silly. Indeed, Figure 3b is even less informative, we know what a 100% AUROC looks like, and it needs no visual assistance. Though Figure 3c contains something for us to learn, it is a weak statement about the role of shuffling the AGORA parameters. Altogether – I would say remove 3a and 3b altogether and change 3c or put it in a supplement.

This was just for consistency of presentation. We have now removed the other cases and simplified the data presented by removing some of the subfigures.

– Regarding Figure 4a: the use of bar plots is discouraged, using box plots would be a lot more informative in the same presentation style and space. 4b: why is the AUROC not stated? The classifier seems hardly impressive.

Done and done.